# Biohybrid Robotic Hand to Investigate Tactile Encoding and Sensorimotor Integration

**DOI:** 10.3390/biomimetics9020078

**Published:** 2024-01-27

**Authors:** Craig Ades, Moaed A. Abd, Douglas T. Hutchinson, Emmanuelle Tognoli, E Du, Jianning Wei, Erik D. Engeberg

**Affiliations:** 1Department of Ocean and Mechanical Engineering, Florida Atlantic University, Boca Raton, FL 33431, USA; cades@fau.edu (C.A.); mabd2015@fau.edu (M.A.A.); edu@fau.edu (E.D.); 2Department of Orthopedics, University of Utah, Salt Lake City, UT 84112, USA; douglas.hutchinson@hsc.utah.edu; 3Center for Complex Systems and Brain Sciences, Florida Atlantic University, Boca Raton, FL 33431, USAjwei@health.fau.edu (J.W.); 4Department of Biomedical Engineering, Florida Atlantic University, Boca Raton, FL 33431, USA; 5Department of Biomedical Science, Charles E. Schmidt College of Medicine, Florida Atlantic University, Boca Raton, FL 33431, USA

**Keywords:** multichannel microelectrode array, amputee, robot, neural network, neuron, biohybrid, prosthetic hand

## Abstract

For people who have experienced a spinal cord injury or an amputation, the recovery of sensation and motor control could be incomplete despite noteworthy advances with invasive neural interfaces. Our objective is to explore the feasibility of a novel biohybrid robotic hand model to investigate aspects of tactile sensation and sensorimotor integration with a pre-clinical research platform. Our new biohybrid model couples an artificial hand with biological neural networks (BNN) cultured in a multichannel microelectrode array (MEA). We decoded neural activity to control a finger of the artificial hand that was outfitted with a tactile sensor. The fingertip sensations were encoded into rapidly adapting (RA) or slowly adapting (SA) mechanoreceptor firing patterns that were used to electrically stimulate the BNN. We classified the coherence between afferent and efferent electrodes in the MEA with a convolutional neural network (CNN) using a transfer learning approach. The BNN exhibited the capacity for functional specialization with the RA and SA patterns, represented by significantly different robotic behavior of the biohybrid hand with respect to the tactile encoding method. Furthermore, the CNN was able to distinguish between RA and SA encoding methods with 97.84% ± 0.65% accuracy when the BNN was provided tactile feedback, averaged across three days in vitro (DIV). This novel biohybrid research platform demonstrates that BNNs are sensitive to tactile encoding methods and can integrate robotic tactile sensations with the motor control of an artificial hand. This opens the possibility of using biohybrid research platforms in the future to study aspects of neural interfaces with minimal human risk.

## 1. Introduction

Brain–machine interfaces (BMIs) have tremendous potential to help disabled people, such as those suffering from a spinal cord injury (SCI) or amputation of a limb [1]. In some cases of SCI, no intact pathway to the peripheral nervous system remains, leaving the affected limbs immobile and insensate. To remedy these kinds of debilitating conditions, invasive BMIs have been used to connect the brain with assistive devices to enable sensation and motor control. However, restoring motor control and sensation from an assistive device in a nearly natural manner remains a scientific holy grail due to the complexity of the problem [2,3]. Nevertheless, noteworthy progress has been made toward this goal. Regarding motor control, a tetraplegic woman who had electrode microarrays implanted in her motor cortex demonstrated ten-dimensional control of a robotic arm [4]. On the sensory feedback side, electrocorticogram electrodes have been used to stimulate the sensory region of the brain corresponding to the hand and lip regions. Two subjects were able to distinguish between different binary pairs of stimulation amplitudes and frequencies; however, only three different frequencies were presented due to limited experimental time [5]. In another study, a tetraplegic subject had microelectrodes implanted in his somatosensory cortex through which he was able to perceive different levels of pressure that were nearly linearly mapped to stimulation amplitude [6]. Research on motor and sensory BMIs has coalesced, yielding a bidirectional interface for a person with tetraplegia who was able to ‘feel’ tactile sensations from a robotic hand via intracortical stimulation of the somatosensory cortex, which helped to improve grasp performance during object manipulation tasks [7].

However, due to the nonminimal risk associated with research on invasive neural interfaces, pre-clinical models are being explored [8]. With this approach, an appropriate animal or in vitro model is chosen to mimic the invasive human intervention as closely as possible. In fact, there is significant interest in establishing pre-clinical assessment tools for neurorobotic platforms such as the exoskeleton of a disabled rat to explore optimal neurorehabilitation strategies after SCI [9]. Efforts such as these can provide a valuable testbed to assist the translation of research findings from the lab to the clinic, like the biohybrid regenerative bioelectronic approach to restore a rat’s amputated peripheral nerve [10]. Another group demonstrated the very large-scale circuit implementation of a multi-scale motor neuron-muscle system to control a human cadaveric hand [11]. The non-deterministic nature of the system enabled behavioral traits to emerge and mimic their natural counterparts, providing a realistic testbed to study the healthy and pathological neuromuscular function of the human hand [12].

In vitro models have also been explored for their potential to provide insights into complex biological systems [13]. Electrically coupling BNNs cultured in MEAs to external robotic systems provides rich potential to study the biological integration of sensation and action in the context of biomedical robotics [14]. Potter and colleagues fruitfully explored the control and robotic embodiment of BNNs in closed-loop architectures with cortical neurons cultured in MEAs [15,16,17]. A breakthrough work with BNNs integrated action and perception by mapping robotic sensory input to stimulate and alter neuronal dynamics, and subsequently impact robotic behavior [18]. Compartmentalizing the MEA chamber into two sections produced different system-level dynamics due to better separation between the efferent and afferent signals [19]. More recently, BNNs cultured in high-density MEAs have been shown to learn new behaviors by providing structured electrical stimulation under the hypothesis that the BNNs would attempt to avoid internal states associated with unpredictable electrical feedback [20]. Another group has presented a brain-on-a-chip model comprised of cortical neurons cultured on an MEA [21]. However, there are no prior works investigating how cortical neurons could perceive robotic sensations of touch in vitro. This could eventually lead to a better understanding of the complex sensation of touch, which is necessary for refined control of the hand. Ultimately, this could improve our ability to restore the capacity for dexterous manual control to amputees and SCI victims.

Severance of afferent tactile pathways deprives amputees of the rich multimodal sensations of touch afforded by the broad distribution of mechanoreceptors in the human fingertips [22], adversely impacting motor control of prosthetic limbs [23,24]. SA mechanoreceptors in human fingertips can detect static pressure; loss of this sensation leaves amputees unaware of the grip force they apply to grasped objects. RA mechanoreceptors are used to sense slipping, contact, and high-frequency vibration [22]; deprivation of this sensation renders amputees unconscious of when grasped objects are slipping [25]. Most prosthetic hands do not have tactile sensors, though there have been strong advances in the field [26,27] for slip detection [28] and pressure sensing [29], among other modalities. The cumulative effect of losing these and other sensations makes prosthetic hands not only difficult to control but also feeling like unnatural extensions of the body with limited capacity for human interaction. By electrically stimulating amputees’ residual peripheral nerves or the somatosensory cortex of tetraplegic people, sensations of touch from assistive devices could be provided to these disabled people. However, one major challenge for conveying tactile sensations through neural interfaces is the mapping from the tactile sensor to the electrical stimulation parameters. Prosthetic fingertip sensations have been encoded in numerous different ways, which impacts not only their qualitative perception but also the closed-loop grasp control performance [30]. Several prior works have investigated frequency [31] or amplitude-modulated spike trains [32] to convey graded biomimetic sensations of touch to amputees outfit with neuroprosthetic limbs interfaced with peripheral nerves [33]. However, there remains much to learn in the field of neuroprosthetics [34] because regulatory, ethical, and financial constraints remain considerable challenges for experimentation in vivo. For these reasons, a limited number of amputees have used bidirectional neuroprosthetic hands thus far [35], and the number of research efforts to explore the impact that different electrical stimulation encoding methods for tactile feedback have upon motor control are fewer still [32,33], which poses a bottleneck to research progress. The same is true for tetraplegic individuals and intracortical microstimulation.

The overarching hypothesis of this paper is that a biohybrid artificial hand model could be useful in studying optimal methods to enable the dexterous control of artificial hands for amputees and SCI victims (Figure 1). We seek to discover whether BNNs are sensitive to different tactile sensation encoding paradigms since these are still actively researched areas with human subjects. Our first hypothesis is that the BNN will integrate tactile feedback with motor control of the hand, which will be classified with a CNN using the coherence between the BNN’s efferent control and afferent feedback signals. Our second hypothesis is that the neurally controlled robotic behavior will be affected by the tactile sensation encoding method that is used to electrically stimulate the BNN. To test these hypotheses, tactile sensations from the robotic fingertip (Figure 1d) are used to biomimetically stimulate the neurons in the MEA (Figure 1a, red) with an RA or SA encoding model. The evoked neuronal activity recorded from the efferent electrode (Figure 1a, black) is decoded to control the robotic hand (Figure 1b–d). We have recently shown that this biohybrid hand can respond to different tactile sensations [36]. New in this work, we delve into the impact that different system embodiment configurations and tactile sensation encoding models have upon the neuronal dynamics of the biohybrid hand. Three afferent and efferent signal combinations were explored for the BNN in the MEA to produce three different embodiment configurations: closed loop (CL), afferent deprived (AD), and efferent substitution (ES). In the CL embodiment configuration, the recording (efferent) electrode was used to control the robotic hand and tactile sensations from the fingertip were used to electrically stimulate the BNN at the afferent electrode, with the neurons in the MEA closing the loop. In the AD embodiment, the BNN was deprived of tactile feedback, analogous to the amputation of a hand after which the amputee is unaware of the fingertip sensations from their prosthesis. In the ES embodiment, the BNN received tactile feedback but the efferent control signal for the robotic hand was decoupled from the BNN activity. Another new contribution in this work is that we probe the capacity of the biohybrid hand model to embody the RA or SA tactile encoding methods with each embodiment configuration (CL, AD, and ES). This is accomplished with a new method to represent the coherence between the stimulating (afferent) and recording (efferent) electrodes within the MEA (Figure 1h) as time-frequency images (TFIs) with a wavelet transform (Figure 1i). Using a transfer learning approach, the resulting images from each neurotactile event are used to train a CNN [37] to classify the mechanoreceptor model (RA, SA) used to electrically stimulate the neurons in each different embodiment configuration (AD, ES, and CL), (Figure 1j). We show that the robotic and neuronal behavior of this biohybrid neuroprosthetic hand model is sensitive to different neural stimulation encoding methods (RA or SA) in the CL configuration. These findings demonstrate the potential for the future use of pre-clinical biohybrid models to investigate human neural interfaces.

## 2. Materials and Methods

This section will give an overview of the biohybrid neuroprosthetic research platform (Section 2.1) and the methods for efferent decoding (Section 2.2) and afferent encoding (Section 2.3) of action potentials from the BNN cultured in the MEA. We next describe the procedures for culturing the BNNs in the MEA (Section 2.4), the protocol for recording each embodiment session (Section 2.5), and how we prevented stimulus crosstalk in the BNN (Section 2.6). We then explain the experimental scenarios for each of the embodiment sessions (Section 2.7) and the effect of the SA or RA stimulus pattern on neurorobotic behavior (Section 2.8). Our pipeline for spatiotemporal analysis consists of a continuous wavelet transform (CWT) for neurotactile event detection (Section 2.9), wavelet coherence (Section 2.10) for observing our 5 min embodiment sessions (macro-scale view, (Section 2.11)), and coherence of individual neurotactile events (micro-scale view (Section 2.12)). The TFIs of these neurotactile events were used to train a CNN with a transfer learning approach for the classification of afferent–efferent coherence within the BNN corresponding to RA and SA encoding methods for each embodiment configuration (Section 2.13).

### 2.1. Biohybrid Neuroprosthetic Hand System Overview

Two subsystems comprise this CL platform. The first subsystem is the robotic unit (Figure 2, right panel), which includes a Shadow Hand [38] (Shadow Robot Company, London, UK) fit with a BioTac SP tactile sensor array (Figure 2g,h) [39] (SynTouch, Montrose, CA, USA), and a computer (Figure 2f) using Robot Operating System (ROS) to manage efferent (Figure 2e, ROS Node 1) and afferent (Figure 2i, ROS Node 2) computations. Node 1 uses efferent signaling from the BNN (Figure 2a–c) to compute the motor control signals for the artificial hand. Node 2 converts the BioTac SP pressure signals into afferent pulse trains of action potentials in real time (Figure 2h,i). The pulse trains were passed to a custom-fabricated action potential generator board (Figure 2j) that triggered electrical stimulations according to a neurocomputational model of haptic encoding, as discussed in Section 2.3.

The second subsystem is the neurophysiological unit (Figure 2, left panel), which includes the head stage of the MEA (MultiChannel Systems, Reutlingen, Germany) that houses the BNN culture, signal collector unit (Figure 2l), interface board (Figure 2b), and their interconnection. The MEA has a 200 µm inter-electrode distance and 30 µm diameter electrodes made from titanium nitride [40]. The data acquisition of the MEA relies on a dedicated high-performance workstation optimized for storage capacity, fast data transfer, and temporal accuracy of the high-density, high-frequency signals (60 channels, 20 kHz) that were sampled from the neuronal culture. Online data were filtered with lowpass and highpass Butterworth filters with cutoff frequencies set to 3.5 kHz and 100 Hz, respectively, to reliably eliminate unstable baselines in real time and avoid artifacts. An additional data stream with the high pass filter set to 1 Hz was preserved for further offline classification with a CNN.

In the present work, a pair of electrodes were selected for their healthy spontaneous activity (Appendix A). One of the sites was assigned the function of recording electrode (Figure 2a): it was the source of efferent signals to control the artificial hand (in the CL and AD configurations). Its spike events were routed through ROS node 1 (Figure 2e) to elicit the robotic finger-tapping behavior. The afferent electrode (Figure 2m) was assigned to be the stimulation electrode, and it used the tactile feedback from the encoded robotic fingertip sensations to stimulate the BNN (in the CL and ES configurations). The tactile signals sensed by the BioTac SP were transformed into afferent trains of action potentials (Figure 1e–g and Figure 2g–i) via the Izhikevich model [41].

The SCB-68A DAQ (National Instruments, Austin, TX, USA) was placed at the interface between both subsystems (Figure 2c). Real-time feedback tests were performed to quantify the delay between instructed stimulation in Simulink (ROS Node 2), its detection by the MultiChannel System’s proprietary software of the MEA, and the return of the signal back to Simulink. CL latency was consistently between 0.8 and 1.0 ms (Appendix A).

### 2.2. Neural Decoding for Robotic Control

In both the CL and AD embodiment configurations, the efferent neurorobotic control signal (*MEA_out_*) was based on spike trains from the recording site of the MEA (Figure 2b) to specify the desired joint angle ($\theta_{D}$) of the Shadow Hand’s index finger metacarpophalangeal joint. The algorithm performed three functions: first, thresholding of its biological neural input signal separated spikes from background noise. Second, a temporal aggregation of the spikes was binned over a neurophysiologically meaningful time interval; and third, the desired index finger joint angle ($\theta_{D}$) control signal was generated for driving robotic finger tapping motions, producing fingertip forces.

We used the extracellular multiunit MEA activity, *V_MEA_* (Figure 2a), from the selected recording electrode as an input to the efferent decoding algorithm for motion control of the Shadow Hand. Spikes (*S*) were detected as:(1)$$S=\left\{\begin{array}{c} 0\;if\;\left(V_{MEA}<V_{thres}\right)\\ 1\;if\;\left(V_{MEA}\;\geq \;V_{thres}\right) \end{array}\right.\;$$
where *V_thresh_* is the action potential spike detection threshold. Subsequently, S is summed over a window of time, *BinSize* (50 ms, (Figure 1b,c)), and compared to a spatiotemporal aggregation coefficient, *S_thres_* (3 spikes to provide a finger tapping rate within the operational bandwidth of the finger). Then, the algorithm outputs a TTL pulse of 100 ms, *MEA_out_*, determined by:(2)$${MEA}_{out}=\left\{\begin{array}{c} \;0\;if\;\left(\sum_{i=0}^{BinSize}S\right)<S_{thres}\\ \;1\;if\;\left(\sum_{i=0}^{BinSize}S\right)\;\geq \;S_{thres} \end{array}\right.$$
*MEA_out_* affects the desired joint angle ($\theta_{D}$) of the Shadow Hand index finger:(3)$$\theta_{D}=\left\{\begin{array}{c} {\;\theta }_{1}\;if\;({MEA}_{out}=0)\\ \;\theta_{2}\;if\;({MEA}_{out}=1) \end{array}\right.$$

Desired joint angle $\theta_{1}$ corresponds to an open hand, where the fingertip does not contact anything (Figure 2g). However, desired joint angle $\theta_{2}$ corresponds to index finger flexion to create fingertip contact with a surface, producing tactile forces (Figure 2h). The measured joint angle, θ, is realized by a PID joint angle controller of the tendon-driven Shadow Hand. The purpose of the PID controller is to minimize the difference between the desired and measured joint angles, which is significant to enable reliable robotic performance across the different experimental conditions explored in this paper. The joint angle (θ) is related to the fingertip force ($F_{DC}$) by:(4)$$B\left(\theta \right)\ddot{\theta }+C\left(\theta ,\dot{\theta }\right)\dot{\theta }+F_{fr}\dot{\theta }+g\left(\theta \right)=\tau_{a}\tau_{fr}-J^{T}\left(\theta \right)F_{DC}$$
where $B\left(\theta \right)$ is the inertia matrix, $C\left(\theta ,\dot{\theta }\right)$ is the matrix representing the Coriolis and centrifugal forces, $F_{fr}$ is the viscous friction coefficient, $g\left(\theta \right)$ is the vector representing the gravitational effect, $\tau_{a}$ is the actuating joint torque, $\tau_{fr}$ is the joint friction, J is the Jacobian matrix of the finger, and $F_{DC}$ is the contact force at the fingertip [42]. Both $F_{DC}$ and the rate of change of the fingertip force ($F_{AC}$) are used within the Izhikevich neurocomputational model to generate SA and RA afferent action potential pulse trains that electrically stimulate the BNN (Figure 1e–g).

An efferent control switch was introduced (Figure 2d) to toggle between the different embodiment modes (CL, AD, and ES). In both the CL and AD embodiment configurations, $\theta_{D}$ was calculated using Equations (1)–(3). However, in the ES configuration, $\theta_{D}$ was decoupled from BNN activity (discussed further in Section 2.7).

### 2.3. Neural Encoding of Robotic Touch Sensations

The afferent neurorobotic feedback signals were implemented in ROS Node 2 (Figure 2i) for SA or RA encodings of tactile sensations as the feedback signals to the MEA (Figure 2h–m). An afferent control switch was introduced (Figure 2k) to control the flow of this afferent signaling during three embodiment modes (CL, AD, and ES). When the afferent switch was closed (CL or ES embodiment configurations), the afferent signal, *MEA_in_* was transmitted to the MEA’s stimulator control unit to stimulate the afferent electrode site. The stimulation wave shape and amplitude were a positive-first biphasic pulse. When the afferent switch was open (AD embodiment, Figure 2k), the afferent signal was not transmitted and no stimulation occurred.

Upon robotic fingertip contact with the environment, the Izhikevich neurocomputational model [41] was employed to convert the tactile fingertip forces ($F_{DC}$, $F_{AC}$, (4)) into spike trains of action potentials representative of RA and SA mechanoreceptors [22] (sample data shown in Figure 1e–g). The neuron input current, I_input_, was generated corresponding to the SA and RA experiments, respectively:(5)$$I_{input_{SA}}=\beta +k_{SA}F_{DC}+k_{RA}F_{AC}$$
(6)$$I_{{input}_{RA}}=\alpha +k_{RA}F_{AC}$$
where, *α*, *β*, $k_{SA}$, and $k_{RA}$ are constants that were chosen to produce physiologically meaningful and distinctly different SA and RA firing rate patterns that mimic different mechanoreceptors. The SA model was responsive to both the steady state force (*F_DC_*) and the force rate of change (*F_AC_*), (5), whereas the RA model was dependent solely upon *F_AC_* (6) [22]. SA and RA pulse trains using input currents (5) and (6), respectively, were generated using the Izhikevich neuron model [41]:(7)$$\dot{v}=Xv^{2}+Yv+Z+I_{input}-u$$
(8)$$\dot{u}=a\left(bv-u\right)$$
(9)if v≥30 mV,thenv←cu←u+dMEAin=4.5 V elsev←vu←uMEAin=0 V
where v is the membrane potential, u is the adaptation variable, and *X*, *Y*, and *Z* are parameters for the model. Parameters *a*, *b*, *c*, and *d* are decay rates, sensitivity, membrane rest potential, and reset values, respectively (Table 1), which are chosen based on [41]. *MEA_in_* is the signal sent to trigger stimulation of the BNN in the MEA system (Figure 2i–m). This model was implemented in real time using Python 3 and ROS Node 2 (Figure 2i) published spike events to the action potential generator board (Figure 2j).

For each event, the custom-made action potential generator board emitted electrical stimuli representing the RA and SA spiking patterns to the MEA. Stimulation pulses are output from the board digitally using the onboard Teensy 3.6 microcontroller via two quad 16-bit digital-to-analog converters with a high-speed SPI interface.

### 2.4. Culturing Biological Neural Networks in Multielectrode Arrays

Primary cortical neurons were harvested from postnatal day 0–1 C57BL/6J mouse pups. All animal procedures were approved by the Institutional Animal Care and Use Committee at Florida Atlantic University and in compliance with the National Institutes of Health Guidelines for the Care and Use of Laboratory Animals. Pups were euthanized by quick decapitation and brains were immediately removed and placed in an ice-cold dissection medium (1 mM sodium pyruvate, 0.1% glucose, 10 mM HEPES, 1% penicillin/streptomycin in HEPES-buffered saline solution). The cortex was extracted under a dissecting microscope and pooled together. The tissue was digested with 0.25% trypsin in the dissection buffer for 15 min at 37 °C followed by further incubation with 0.04% DNase I (Sigma-Aldrich, St. Louis, MO, USA) for 5 min at room temperature. The digested tissue was triturated with a fire-polished glass pipette 10 times and cells were pelleted by centrifugation. Cells were counted using the Trypan blue exclusion method with an automated LUNA-II^TM^ cell counter (Logos Biosystem, Annandale, VA, USA) and plated at ~5000 cells/mm^2^ in a polyethylenimine-coated MEA-60 chamber (Figure 1a, Appendix A). The culture was maintained in the BrainPhys^TM^ neuronal culture medium (STEMCELL Technologies Inc., Vancouver, BC, Canada) supplemented with 2% B27 (Invitrogen, Waltham, MA, USA) and 1% GlutaMAX^TM^ (Invitrogen). Culture media in the chamber were half-changed every three days. Spontaneous structural and functional connectivity was allowed to mature before the biohybrid neurorobotic experiments.

### 2.5. MEA Experimental Recording Protocol

Each day of experiments began with verification that system noise was within acceptable bounds (10–20 µV) with a fixed-resistance test chamber provided by Multichannel Systems. The test chamber was recorded for 1 min outside the incubator and then placed inside the incubator, allowing 20 min for the temperature to stabilize. Confirmation data were subsequently recorded for 1 min. After confirming the system noise level was stable and consistently within the acceptable range of 10–20 µV on each DIV, the MEA chamber containing the BNN was inserted into the headstage (Figure 2) and allowed to settle for 5 min. The chamber with the BNN was recorded for 5 min with no stimulation to obtain a baseline of spontaneous neural activity. Datasets were collected over 3 DIV. There were 3 embodiment sessions per day (CL, AD, and ES), and 2 mechanoreceptor firing pattern encodings per embodiment session (SA and RA), totaling 18 datasets over 3 DIV.

### 2.6. Neurons Close the Loop: Preventing Stimulation Crosstalk

To study the BNN in the MEA culture, it was important to ensure that the afferent stimulation (*MEA_in_*, Figure 2m) did not propagate through the culture medium to the recording electrode (*MEA_out_*, Figure 2a) and cause depolarization of the efferent neuronal population directly. Therefore, we chose the activation threshold (*V_thresh_*) for *V_MEA_* (1) to be higher than the observed crosstalk from the stimulation electrode. We verified that no temporally coincident spiking activity exceeded the background noise level at the stimulation site’s 8 neighboring electrodes, showing a minimum of full width, half amplitude drop. In this way, we ensured that electrical activity from *V_MEA_* and *MEA_out_* (Figure 2a,b) was due to the synaptic connections between the recording (Figure 2a) and stimulation (Figure 2m) electrodes in the MEA chamber, not due to direct stimulation from a distance.

### 2.7. Biological Neural Network Robotic Embodiment Sessions

In the CL embodiment sessions, both the afferent and efferent switches (Figure 2d,k) were toggled to allow bidirectional communication between the BNN and the robotic system. This allowed the decoded neural signals to be coupled to the tactile encodings (RA and SA) through the dynamics of the BNN’s efferent and afferent electrodes. Appendix A shows the CL embodiment configuration with the SA encoding while Appendix A shows the CL configuration with the RA encoding. In both tactile encoding methods, neural activity at the efferent electrode was used to control the finger motion using Equations (1)–(3). However, the SA encoding method used Equation (5) while the RA encoding used Equation (6) to produce distinctly different patterns of electrical stimulation for tactile feedback at the afferent electrode in the BNN.

In the AD sessions, the efferent control signal was again formulated using Equations (1)–(3); however, the afferent pathway was disconnected by toggling the afferent switch (Figure 2k) to an open-circuit state (no stimulation of tactile sensory signals to the BNN). This provided a mismatch between the motor output and the sensory inputs, decoupling the interactions for analysis of the biohybrid robotic behavior due to spontaneous BNN activity. It should be noted that an analogy can be made between this AD embodiment configuration and traditional control of a prosthetic hand by an upper-limb-absent person: the person can control the hand but has no awareness of touch sensations from the hand.

In the ES sessions, the afferent switch (Figure 2k) was in a closed-circuit state allowing tactile feedback to the BNN; however, the efferent switch (Figure 2d) was toggled to substitute the decoded motor commands from the MEA with a fixed stimulation pattern to control the movement of the finger. For these ES embodiment sessions, we enabled movement of the finger by programming the desired finger joint angle ($\theta_{D}$) to be a 0.25 Hz square wave with a lower amplitude of $\theta_{1}$ and upper amplitude of $\theta_{2}$, which repetitively produced 2 s of tactile contact with the environment and 2 s without contact. Thus, the motor output and the sensory inputs were independent in the ES configuration.

### 2.8. Effect of Mechanoreceptor Encoding Model on Neurorobotic Behavior

To investigate the capacity for functional specialization of the biohybrid neuroprosthetic hand model, we analyzed the inter-tap-interval (ITI) of the fingertip for both SA and RA experiments with all three embodiment configurations (CL, AD, and ES). The ITI represents the time interval between finger taps initiated by the neural activity. We extracted the timestamps of each neurotactile event and calculated the time interval between each event, represented as the ITI (Figure 3(Aa,Ab)).

For each embodiment session (CL, AD, and ES), MATLAB was used to perform an unbalanced ANOVA between the ITIs of each RA and SA encoding method. We selected a *p*-value of 0.05 for statistical significance on each of the three DIVs. We used this to determine if the RA and SA tactile encoding methods significantly impacted the neurorobotic behavior for each of the embodiment sessions, represented through their ITI. This ITI metric is an indicator of whether the BNN can process the two tactile sensation encoding models (RA and SA) differently using closed-loop feedback, which has important implications for the development of a pre-clinical model of neuroprosthetic interfaces.

### 2.9. Continuous Wavelet Transform for Neurotactile Event Detection

For a robust event detector, we chose a CWT to process the efferent (Figure 3(Ag,Bg)) and afferent signals (Figure 3(Aj,Bj)). Data were processed through the CWT using a complex Morlet as the wavelet type for extracting the mean amplitude-squared power (MASP) across time for multiple frequency bands (Figure 3(Ai,Al,Bi,Bl)).

To obtain the MASP, we first calculated the scales for the CWT based on the center frequencies within a designated wavelet range of 100–4000 Hz. Next, the absolute value of the CWT was squared to obtain the amplitude-squared power (ASP) for the selected frequency ranges (Figure 3(Ah,Ak,Bh,Bk)). We scaled ASP (SASP) by dividing all values by the maximum value, obtaining a scaled range of 0 to 1. We further adjusted the range to eliminate artifacts (0.05–0.4). Following this, we took the mean of SASP across time and frequency to obtain MASP (Figure 3(Ai,Al,Bi,Bl)). MASP was smoothed using a moving mean function (movmean in MATLAB) with an averaging window of 50 ms. Finally, we scaled MASP by dividing all values by the maximum value, obtaining a scaled range of 0 to 1. A peak detection algorithm (findpeaks) was used in MATLAB to extract the timestamps of peaks exceeding a threshold of 0.5 with a minimum peak distance of 0.5 s to avoid multiple local maxima.

### 2.10. Efferent–Afferent Coherence to Investigate Spatiotemporal Coordination

We explored the ability of the BNN to exhibit different behaviors during CL (Figure 3(Aa–Af)), AD (Figure 3(Ba–Bf)), and ES embodiment configurations. Wavelet coherence was used to generate TFIs of how the amplitudes of BNN activity at two spatial locations (recording and stimulating electrodes) coordinated at different frequencies over 1 s time scales (Figure 3(Am,Bm)), as well as during the entire 5 min embodiment sessions (Figure 3(Ao,Bo) and Figure 4). This process was completed by first extracting the smoothed MASP for both the afferent and efferent electrodes. For the embodiment sessions containing 5 min of data, the smoothed MASP was downsampled from 20 kHz to 2 kHz to reduce the computational expense while preserving the spectral information needed for further analysis (Figure 3(An,Bn)). For the individual microscale neurotactile events (Figure 3(Am,Bm)), we maintained the 20 kHz sampling rate as it was not computationally expensive.

### 2.11. Macro-Scale View of Robotic Embodiment Session Coherence

For each of the three robotic embodiment sessions per DIV (AD, ES, and CL), MASP coherence between the afferent and efferent sites was extracted. These time-frequency representations were compared with each other to obtain the coherence for the 5 min duration of each embodiment session (Figure 3(Ao,Bo) and Figure 4). Wavelet coherence was calculated with the wcoherence function in MATLAB. A cone of influence was used to eliminate data affected by edge conditions with lower frequencies at smaller timescales. The wavelet used was an analytic Morlet wavelet.

### 2.12. Micro-Scale View of Neurotactile Event Coherence

For each timestamped neurotactile event corresponding to every time the BNN elicited a finger tap (Figure 3(Am,Bm) and Figure 5), a 300 ms segment before and 700 ms segment after each timestamp was extracted for both the afferent and efferent electrodes. This provided a temporally synchronized window before stimulation (efferent motor commands) and for the duration of the stimulation (afferent tactile sensations). For each 1 s time window, the wavelet coherence was again calculated between the efferent and afferent electrodes. These resulting TFIs (Figure 5c,f–h) were used in a transfer learning paradigm to train a CNN [37]. This spatiotemporal coherence analysis allowed us to compare the impact of the tactile encoding patterns (RA and SA) and system configurations (CL, AD, and ES) on each DIV.

### 2.13. Transfer Learning with CNN to Classify BNN Activity

For each robotic embodiment session, we applied transfer learning with a CNN to classify RA and SA neurotactile events for each DIV and every embodiment configuration (AD, ES, CL). Transfer learning is an approach using a pretrained artificial neural network to transfer knowledge from a related task to a new task [43]. We exploit transfer learning of general image recognition with a CNN, to a specific task of TFI classification [37] corresponding to the afferent–efferent coherence of the BNN within the biohybrid neurorobotic hand system (Figure 1j). We retrained the last three layers of AlexNet to enable the network to retain the earlier trained layers for recognizing shape, size, color, etc., and achieve high classification accuracies with only needing between 80 and 150 new images from each session and DIV (Figure 1i). We extracted the TFIs of the neurotactile events for RA and SA on each DIV to train the CNN. Illustrative training data in the transfer learning program are shown for both the RA and SA mechanoreceptor encodings with the CL, AD, and ES system configurations (Figure 5c,f–h). The datasets were split into training (70%) and validation (30%) portions. We loaded the pretrained network and configured the input size of the images to be 227 × 227 × 3. We replaced the last 3 layers with a fully connected layer, a softmax layer, and a classification output layer. We then specified additional augmentation operations to perform on the training images: randomly flipping the training images along the vertical axis and randomly translating them up to 30 pixels horizontally and vertically. Data augmentation helped prevent the network from overfitting and memorizing the exact details of the training images. Next, the network was trained for 10 epochs and the results of the final 10 iterations were averaged.

For each DIV, an ANOVA was performed to determine if there was a statistically significant difference between the classification accuracies to distinguish between RA and SA tactile encoding methods used with the three embodiment configurations (CL, AD, and ES).

## 3. Results

Both the mechanoreceptor encoding model (RA, SA) and the embodiment configuration affected the biohybrid hand behavior. We begin describing these impacts by contrasting the CL and AD configurations with the SA encoding method. In the CL configuration with the SA mechanoreceptor encoding (Appendix A), transfer of information through the biohybrid neuroprosthetic platform began when MEA efferent site activity (*V_MEA_*, (Figure 3(Aa)) rose above the voltage threshold, *V_thres_*_,_ to trigger a spike (*S*, Figure 3(Ab)). When *S* was triggered *S_thres_* = 3 times within *BinSize* = 50 ms (2) to trigger *MEA_out_*, the desired joint angle of the finger ($\theta_{D}$) increased from ${\;\theta }_{1}$ to ${\;\theta }_{2}$ (3). This caused the finger joint PID controller to increase the joint angle (θ, Figure 3(Ac)) so that the fingertip contacted the environment, increasing the fingertip force (*F_DC_*, (4)), (Figure 3(Ad)) and force rate of change (*F_AC_*, Figure 3(Ae)). The SA encoding of fingertip forces (5) produced spike trains that were used to stimulate *MEA_in_* during the CL experiments (Figure 3(Af)). This process repeated, producing repetitive finger-tapping behavior with variable ITI dependent upon the afferent tactile encoding (RA and SA) and embodiment configuration (CL, AD, and ES). One illustrative efferent event (Figure 3(Ag)) had power components in the 100 Hz–4 kHz spectrum (Figure 3(Ah)). The resultant finger tap-evoked afferent event (Figure 3(Aj)) also had significant power components in the 100 Hz–4 kHz spectrum (Figure 3(Ak)). The coherence between the efferent and afferent events showed substantial power in the 100 Hz–4 kHz spectrum (Figure 3(Am)).

However, the neurorobotic behavior in the AD system configuration was quite different because there was no feedback to the BNN. Activity at the recording (efferent) electrode (Figure 3(Ba)) did elicit spikes (*S*, Figure 3(Bb)) that drove the finger tapping motion (Figure 3(Bc)), producing increases in fingertip forces *F_DC_* (Figure 3(Bd)) and *F_AC_* (Figure 3(Be)). One illustrative AD efferent event (Figure 3(Bg)) had power broadly in the 100 Hz-4 kHz spectrum (Figure 3(Bh)), as did activity at the afferent electrode (Figure 3(Bk)). However, activity at the afferent electrode (Figure 3(Bf,Bj)) did not exhibit significant temporal coupling with the efferent electrode and the coherence between the efferent and afferent events had much less power (Figure 3(Bm)) in comparison to the CL configuration (Figure 3(Am)).

Looking at the macroscale 5 min embodiment sessions, the CL configuration also had substantially more coherence in the low-frequency bands (Figure 3(Ao)) than the AD configuration (Figure 3(Bo)) for both the RA and SA encodings (Figure 4).

Contrasting the RA and SA encoding models with each embodiment configuration also revealed that the CL configuration consistently produced starkly different TFIs of efferent–afferent coherence based on the mechanoreceptor encoding model (See also Appendix A). Illustrative efferent data from the CL embodiment configuration drove a finger tap (Figure 5a). The SA mechanoreceptor encoding model produced resultant stimulation at the afferent electrode (Figure 5b) that produced a TFI with a pattern of strong spatiotemporal coherence (Figure 5c). However, the RA encoding produced a noticeably different pattern of coherence in the CL configuration (Figure 5d–f)—most of the coherence coincided with the beginning and end of the finger tap. This is likely because the RA model (6) is dependent only upon the dynamic force *F_AC_*, whereas the SA model (5) is based on both *F_AC_* and the steady state force, *F_DC_*. Illustrative TFIs in each embodiment configuration revealed inconsistent coherence patterns with the AD and ES configurations in both the SA (Figure 5g) and RA encodings (Figure 5h). In the AD configuration, coherence was minimal because there was no feedback to the MEA. But in the ES configuration, there were occasional times with strong coherence likely due to temporal alignment of the BNN activity at the afferent electrode with the periodic square wave that drove the finger tapping motion in that case.

### 3.1. Mechanoreceptor Encoding Model Impacted CL Biohyrid Hand Behavior

In the CL embodiment configuration, the ITI of the SA encoding method (Figure 3(Aa,Ab)) was significantly longer than it was with the RA encoding on each of the 3 DIVs (*p* < 0.05, Figure 6a). This indicated that the tactile information fed back to the afferent stimulation site consistently altered the BNN behaviors that the robotic hand adopted within the CL system. In other words, the tactile feedback was embodied by the BNN for functional specialization with respect to the mechanoreceptor encoding model (RA and SA) in the CL system configuration.

In the ES sessions, no consistent ITI trend emerged due to the tactile sensation encoding method. On DIV 21, there was no significant difference between RA and SA encodings. On DIV22, the ITI for SA was significantly slower than RA, and on DIV23, it was the opposite (*p* < 0.05). While afferent sensory feedback from the fingertip was provided to the BNN in the ES configuration, the efferent control signal and afferent neural feedback were independent.

In the AD sessions, there was no statistically significant difference in the ITI rates for any DIV with the AD configuration (*p* > 0.05), contrasting starkly with the capacity for functional specialization of the neurorobotic behavior elicited by the tactile encoding methods in the CL configuration (Figure 6a).

### 3.2. Spatiotemporal Coherence Is Impacted by Biohybrid Hand Embodiment Configuration

The CNN was able to classify TFIs of spatiotemporal coherence from RA and SA neurotactile events with high accuracy in the CL embodiment configuration. Accuracy decreased significantly in the ES configuration and plummeted further in the AD configuration (Figure 6b). Averaged across all three DIVs, the mean and standard deviations of the classification accuracy for the AD, ES, and CL embodiment configurations were 51.43% ± 3.03%, 79.20% ± 6.20%, and 97.84% ± 0.65%, respectively. Figure 6c and Figure 6d respectively show illustrative accuracy and loss during the CNN training with the CL embodiment configuration on DIV23. The highest accuracy was obtained via CL coupling with 100% ± 0% on DIV22, while the maximum ES accuracy was 79.83% ± 7.13% on DIV 23. The maximum accuracy for AD was 53.19% ± 3.77% on DIV23 (Figure 6b). The ANOVA showed that there was a statistically significant difference between the classification accuracies with each embodiment configuration on all three DIVs (*p* < 0.01).

## 4. Discussion

The biohybrid neuroprosthetic hand model that we have introduced in this paper is sensitive to the tactile encoding method from the robotic fingertip at the behavioral level as demonstrated by significant differences in ITIs (Figure 6a). Furthermore, we were able to use BNN activity to detect the tactile encoding method in the CL configuration as evidenced by the high classification accuracy of the TFIs corresponding to spatiotemporal coherence between the afferent and efferent electrodes (Figure 6b). In the CL configuration, there was a consistent pattern where the SA encoding method produced slower ITIs than the RA encoding on each DIV (Figure 6a). This pattern was not consistent when the BNN was deprived of tactile feedback in the AD case, nor was it consistent when the efferent control signal was independent of BNN activity in the ES case. A likely explanation is that the CL configuration enabled the neurorobotic system’s capacity for sensorimotor integration using the neuronal processing capability of the BNN. The inconsistent neuronal dynamics from DIV21 to DIV23 occurred only in the ES embodiment configuration in which tactile feedback was given to the BNN but the motor control signal for the hand was based on a pre-determined waveform. The back-and-forth behavior across DIVs with the ES configuration was due to this disconnection between the feedback and control signals that prevented the neuronal processing of the tactile feedback signals from propagating to the robotic controller. This contrasted with the CL case in which the feedback and motor control signals were allowed to be coupled by the neurons in the MEA. In the CL embodiment, the neuronal processing was consistent across each of the three days with respect to the tactile encoding method (RA, SA). This resulted in more consistent behavior across each DIV further evidenced by the significantly higher classification accuracy of the CL TFIs in comparison to the AD and ES configurations (Figure 6b).

Another difference between the system configurations is the stronger presence of low-frequency coherence (<10 Hz) in the CL case relative to the ES and AD configurations that was apparent in the macroscale 5 min time windows (Figure 3(Ao,Bo) and Figure 4). There is evidence to suggest that these slow waves can precede or temporally coincide with spiking events [44]. While it is not possible to recover causal influences from unobserved nodes in a network, there could be some influence of the temporally coupled spiking from the neurotactile events upon the local field potentials [45] at the recording (efferent) and stimulating (afferent) electrodes. The classification of local field potentials has been useful in cognitive neural prosthetics to indicate intracortical processing and motion planning [46]. Reinforced by CL feedback, the low-frequency local field potential signals could be subthreshold extracellular ‘fingerprints’ that predict network activity [47] (Figure 6b). This could be helpful in overcoming some significant challenges that prevent the realization of an ideal neural interface for people, such as the capacity to decode neural activity corresponding to a person’s desired action from an assistive device [48]. One recent idea to overcome this challenge was to use EEG recordings from sensorimotor regions of an upper-limb-absent person’s brain to improve the classification accuracy of electroneurogram recordings for artificial hand control [49]. Another technique used TFIs from surface electromyogram signals to train a CNN to classify human movements [37]. In this paper, we have explored a new concept between these two aforementioned approaches [37,49] by classifying sensorimotor integration using TFIs created from the coherence between efferent and afferent electrodes interfaced with the robotic hand.

In people, the corticospinal system includes direct cortico-motorneuronal connections that operate in parallel with indirect connections [50], such as somatosensory inputs from the hand [51]. Interrupting the sensorimotor feedback loop of the human hand negatively impacts corticomuscular coherence [52,53]. Moreover, both afferent and efferent systems impact corticomuscular coherence in tasks involving human precision grip [53]. In our study, the AD experiments represented an interruption in the sensorimotor feedback loop and also revealed decreased afferent–efferent coherence compared to the ES and CL configurations (Figure 3, Figure 4 and Figure 5), which could be one short-term factor that drives long-term cortical network reorganization [54]. The long-term impact of a traumatic arm or hand amputation is a large-scale cortical reorganization at the network level, extending beyond the sensorimotor cortex [55]. Conversely, approaches such as targeted muscle and sensory restoration for amputees have improved functional connectivity between the primary motor and primary somatosensory cortex [56]. However, this phenomenon is difficult to measure in vivo due to the limited number of subjects and the constraints that cortical imaging studies impose on their daily lives. It could therefore be beneficial to study this phenomenon with a model that is in a more controlled environment, such as with the new biohybrid hand model (Figure 1). This warrants additional investigation in the future, simultaneously using live cell imaging since cortical reorganization can lead not only to phantom limb pain (PLP) after amputation [57] but also to a reduction in PLP [58].

In this paper, we used dissociated cortical neurons from rats in the MEA [59], which have obvious structural differences from the human brain. In the future, it would be fruitful to investigate different biohybrid models coupling artificial hands with brain region-specific organoids [60]. Furthermore, these brain organoids could be assembled with muscle spheroids to model the cortical–spinal–muscle pathway [61]. This kind of assembloid could offer numerous opportunities to study touch perception and grasp control by feeding back tactile sensations to either the spinal or cortical organoids while simultaneously recording muscular activity for efferent control of the artificial hand. We could also expand this technique for a model of the peripheral nervous system [10] for amputees if coupled with a human peripheral nerve on a chip [62].

The current pipeline to restore a single sensation of touch for amputees can be generalized with two key steps: first, perform a basic research study (often invasively), typically with people or non-human primates to gain an understanding of a single aspect of touch perception. Second, apply this knowledge to a disabled person or healthy volunteers. We believe that this approach will be difficult to scale up to fully understand and restore the myriad of distinctly different sensations of touch for object friction, moisture, curvature, temperature, edge perception, pain, etc.

To substantiate this claim, we point to a notable study with non-human primates on the tactile perception of different textures in which it was postulated that coarse textural features are encoded spatially via SA type 1 afferents whereas fine textural features are encoded temporally with RA and Pacinian Corpuscle (PC) afferents [63]. This research was subsequently translated into the field of neuroprosthetics to enable an amputee to distinguish between several different kinds of surface textures by rate-sensitive stimulation through a single electrode in the median nerve [64]. This is an impressive outcome. However, the authors in [64] hypothesized that multi-electrode stimulation could enable more complex texture discrimination tasks. But how many feedback channels should be used? And what combinations of SA, RA, and PC channels will optimize the capacity for texture discrimination? Pre-clinical models such as the work we have presented or other models including brain organoids could be used to explore optimal methods for this kind of functionality, indeed as they already have been for unsupervised learning tasks like speech pattern recognition [65]; for a review, see [66]. We thus advocate for a paradigm shift to explore the capacity of biohybrid models to be incorporated into the neuroprosthetic research pipeline to reduce risks and pain for human subjects, accelerate research progress, and lower costs. Finite element analysis software is commonly used in neuroprosthetic research [64], in a similar vein, so also biohybrid models could be used to aid the process.

Due to the highly subjective nature of perception, it should be noted that biohybrid models will likely not be useful to fine tune the qualia of perceived sensations; rather, they can be used as a testbed to estimate the number of independent sensations that could be perceived for any combination of electrodes, haptic modalities, and stimulation parameters. There will always be a need to customize the perceived quality of sensations with the end users. In fact, a reinforcement learning approach has demonstrated the potential to tailor stimulation parameters like pulse width and amplitude uniquely to individual subjects [67]. However, it is well known that reinforcement learning approaches can require a burdensome number of environmental interactions to learn optimal policies, which highlights the value of a pre-clinical biohybrid model to reduce the complexity of the problem prior to in vivo experimentation.

Many aspects of dexterous grasp control have been studied; however, some facets of dexterous control are not yet completely understood [68]. Biohybrid models could be useful to generate testable hypotheses regarding some of the unknowns like proprioception, how precise grasp forces are encoded in the somatosensory cortex and motor cortex, and how proprioception is integrated with tactile cues to infer object parameters like grasped object size and shape. In the future, we will investigate the use of our biohybrid hand for closed-loop grasp control during increasingly complex object manipulation and environmental interaction tasks [69].

Additional applications where a pre-clinical biohybrid hand testbed could be useful in the near term include the evaluation of novel neural encoding/decoding strategies which are time-consuming and burdensome to quantify with human subjects. Often, sensory encoding and motor decoding algorithms are investigated independently, and their interactive effects are not explored due to limited time with human subjects. A biohybrid hand model could provide a physical testbed to evaluate the interaction between myriad sensory encoding and motor decoding algorithms. Furthermore, the novel testbed could allow novel bioinspired control algorithms to be implemented and evaluated. The tradeoffs of these new algorithms could be directly compared systematically in a controlled way that is difficult to assess with human subjects, enabling the capability for identifying superior algorithms prior to human experimentation. In the future, further advancements may be possible, such as in vitro testing for electrode material biocompatibility [70] with a more realistic model, or assessing optimal conditions for surgical interventions to restore prosthetic limb function [71] with a non-human model [10]. There is significant value in developing realistic pre-clinical models of neural interfaces, which could reduce risks to human subjects, lower costs required to conduct research, democratize access to perform neuroprosthetic research, and diminish the burden of complying with regulatory requirements. Overcoming the hurdles to achieve this goal could profoundly impact millions of disabled people.

## 5. Conclusion

We have developed a new biohybrid neuroprosthetic research platform comprised of a dexterous artificial hand that was electrically interfaced with a BNN. We have shown that the method of encoding tactile fingertip sensations with SA or RA neuromimetic mechanoreceptor models affects the behavior of the biohybrid hand. We have also demonstrated that the system is capable of sensorimotor integration by contrasting CL connectivity of motor control and feedback signals with AD and ES system configurations. The CL system configuration exhibited strong coherence between stimulating (afferent) and recording (efferent) electrodes of the MEA that were respectively interfaced with the tactile sensation and motor control capacities of the artificial hand. Furthermore, we have introduced a new technique to classify sensorimotor integration of neuronal activity by creating TFIs from the wavelet coherence between afferent and efferent activity in the BNN. The resultant TFIs of spatiotemporal coherence were classified by a CNN using a transfer learning approach showing that the tactile encoding method (RA and SA) was easily distinguishable in the CL system configuration. Classification accuracy plummeted in the AD and ES system configurations, demonstrating the capacity for functional specialization of BNN activity relative to the fingertip tactile encoding method with the CL configuration. This biohybrid neuroprosthetic research platform has the potential to become a useful pre-clinical tool to investigate optimal methods for neurorehabilitation with assistive robots.

## Figures and Tables

**Figure 1 biomimetics-09-00078-f001:**
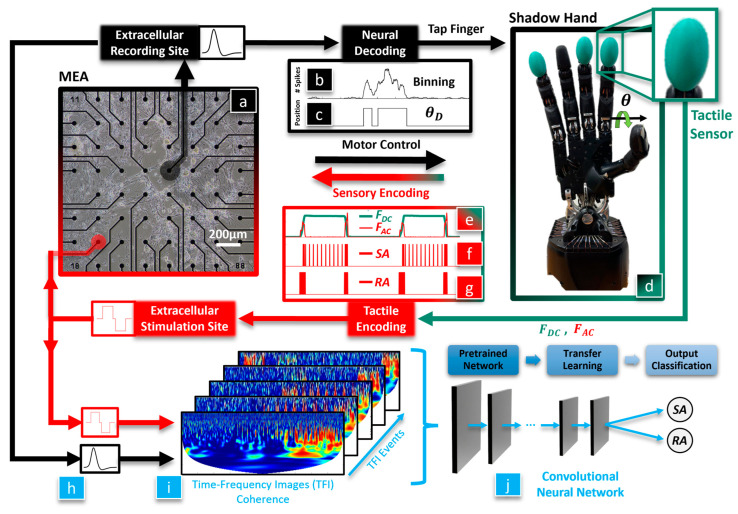
Closed loop biohybrid neuroprosthetic hand model. Within the biological neural network cultured in the MEA, (**a**) neural activity at a recording (efferent) electrode is binned to compute the average firing rate (**b**). The desired joint angle $\theta_{D}$ (**c**) was increased when firing rates above a set threshold were detected to initiate a finger-tapping motion of the Shadow Robot Hand (**d**). Touch sensations were decomposed into steady, *F_DC_*, and transient, *F_AC_*, components (**e**) and then converted into pulse trains using the Izhikevich neuron model (encoding slowly adapting (SA) signals in (**f**) and rapidly adapting (RA) in (**g**)). Those biomimetic afferent pulse trains electrically stimulated the afferent electrode of the MEA. Offline analysis extracts neurotactile events from the recording and stimulation sites (**h**). These are processed through a spatiotemporal wavelet coherence algorithm to produce time-frequency image (TFI) events (**i**). Transfer learning (**j**) was used to retrain the last three layers of a CNN (AlexNet) using the TFIs to classify the mechanoreceptor encoding model (SA or RA) used in each embodiment configuration (CL, AD, ES).

**Figure 2 biomimetics-09-00078-f002:**
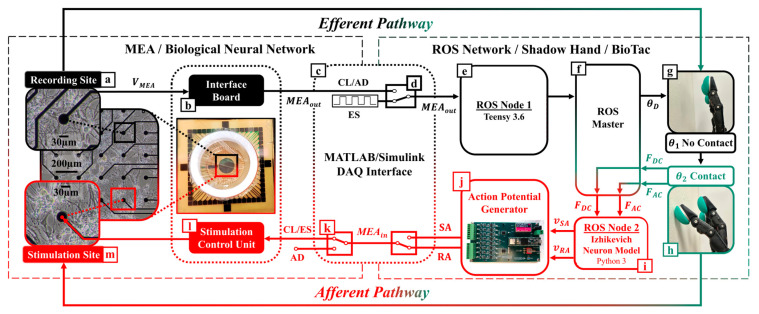
System-level diagram showing the hardware implementation of the biohybrid neuroprosthetic research platform. (**a**) An extracellular recording site depicting MEA spiking activity, V_MEA_, is transmitted to the Multichannel Systems Interface Board (**b**) to detect neural events. For each event, a TTL pulse, *MEA_out_*, is relayed through one of the analog input ports of the DAQ (**c**) to the efferent control switch (**d**), which can toggle between signals input to ROS Node 1 (**e**). ROS Master (**f**) receives the motor control signals for controlling the desired joint angle, *θ**_D_*, and relays them to the Shadow Hand controller. This initiated a tapping motion from not being in contact with the external environment, *θ*_1_ (**g**), to being in contact, *θ*_2_ (**h**). Upon contact, the BioTac tactile sensor (**h**) generates a force, *F_DC_*, and force rate of change, *F_AC_*, that are transmitted to ROS Node 2 (**i**) and converted into rapidly adapting (RA) and slowly adapting (SA) mechanoreceptor firing patterns using the Izhikevich neuron model. SA or RA pulse trains are transmitted to the action potential generator board (**j**), which converts each pulse into a biomimetic action potential waveform. These waveform representations of RA and SA are sent into (**c**), which transmits a signal, *MEA_in_*, through the afferent control switch (**k**), which can be toggled to deprive or enable afferent feedback to the BNN. When enabled, afferent signals pass through the stimulation control unit (**l**) to stimulate the BNN at the afferent electrode with either the RA or SA model to evoke activity at the stimulation site (**m**). The neurons in the MEA close the loop between the recording and stimulation sites.

**Figure 3 biomimetics-09-00078-f003:**
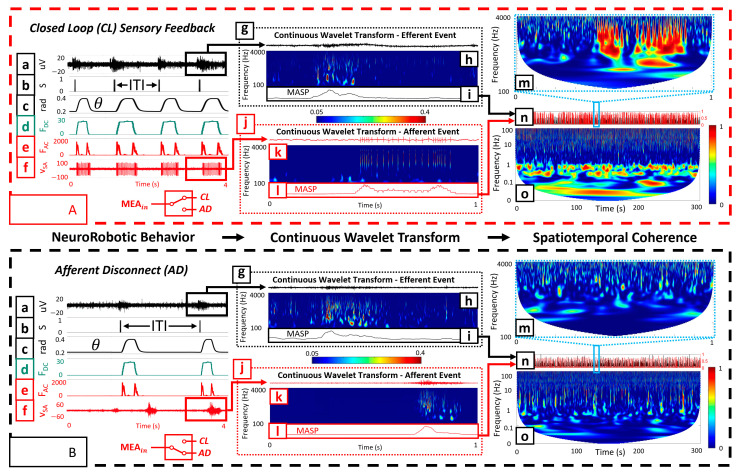
Neurorobotic signal processing pipeline with the CL (**A**) and AD (**B**) embodiment configurations. (**a**) Neural activity at the efferent recording electrode. (**b**) Detected events elicit the tap of a finger and the time in between taps is the inter-tap-interval (ITI). (**c**) The joint angle (θ) varies from a no-contact state to a contact state. (**d**) Upon contact, the static force *F_DC_* and dynamic force *F_AC_* (**e**) are encoded into the (**f**) SA tactile stimulation pattern. (**g**,**j**) Single efferent and afferent events. (**h**,**k**) The amplitude-squared power (ASP) of the efferent and afferent events, respectively. (**i**,**l**) The mean ASP (MASP), is calculated across time. (**m**) Spatiotemporal wavelet coherence is generated for each neurotactile event and (**o**) each 5 min embodiment session using the (**n**) afferent and efferent MASP. There was significantly more coherence in the CL embodiment (**A**) than in the AD embodiment configuration (**B**). Note that no afferent electrical stimulation occurred in the AD configuration, electrical activity recorded in this case is spontaneous neural activity (**Bf**).

**Figure 4 biomimetics-09-00078-f004:**
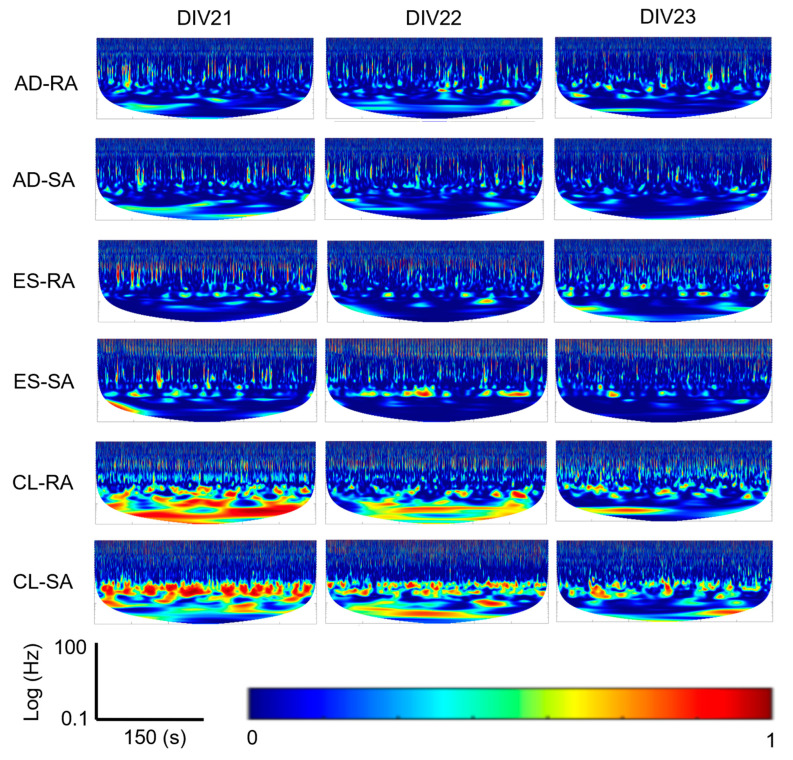
Macroscale view of 5 min embodiment sessions across three days in vitro (DIV) for the afferent-deprived (AD), efferent substitution (ES), and closed-loop (CL) configurations with both the rapidly adapting (RA) and slowly adapting (SA) encoding methods. Data from the CL configurations had significantly more low-frequency power on each DIV.

**Figure 5 biomimetics-09-00078-f005:**
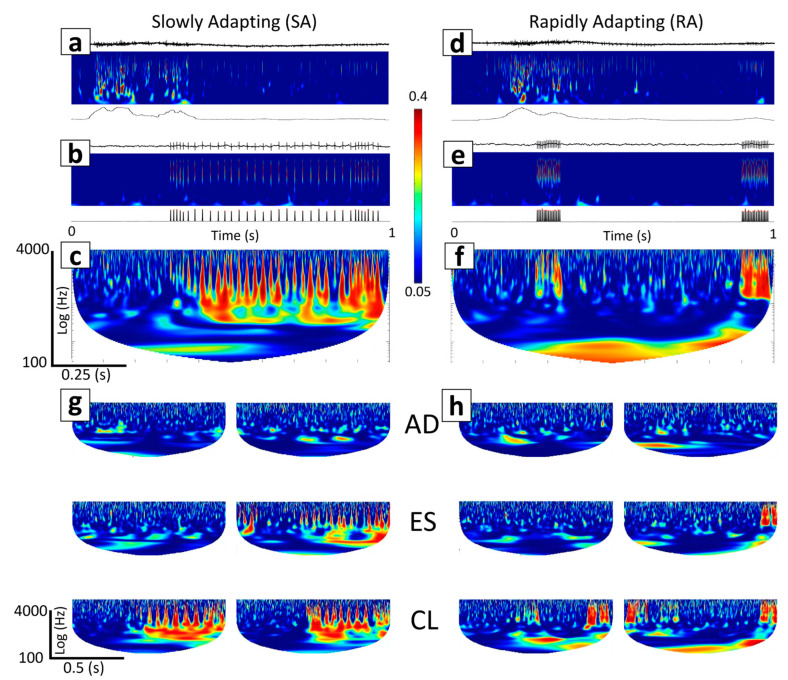
Illustrative neurotactile events with the SA (**left** column) and RA (**right** column) mechanoreceptor encoding methods. (**a**) In the CL embodiment configuration, detected efferent neural activity drove a finger tap. (**b**) The tap of the finger produced the SA firing pattern at the afferent electrode. (**c**) The spatiotemporal coherence between the efferent and afferent electrodes revealed a high correlation. (**d**) Again in the CL configuration, efferent activity drove a finger tap. (**e**) With the RA mechanoreceptor model, the finger tap created the RA firing pattern at the afferent electrode. (**f**) There was strong spatiotemporal coherence between the afferent and efferent electrodes with the RA model that produced a noticeably different pattern than the SA model. (**g**) Six random examples of time-frequency images used to train the CNN with the SA and (**h**) RA models with the AD, ES, and CL embodiment configurations.

**Figure 6 biomimetics-09-00078-f006:**
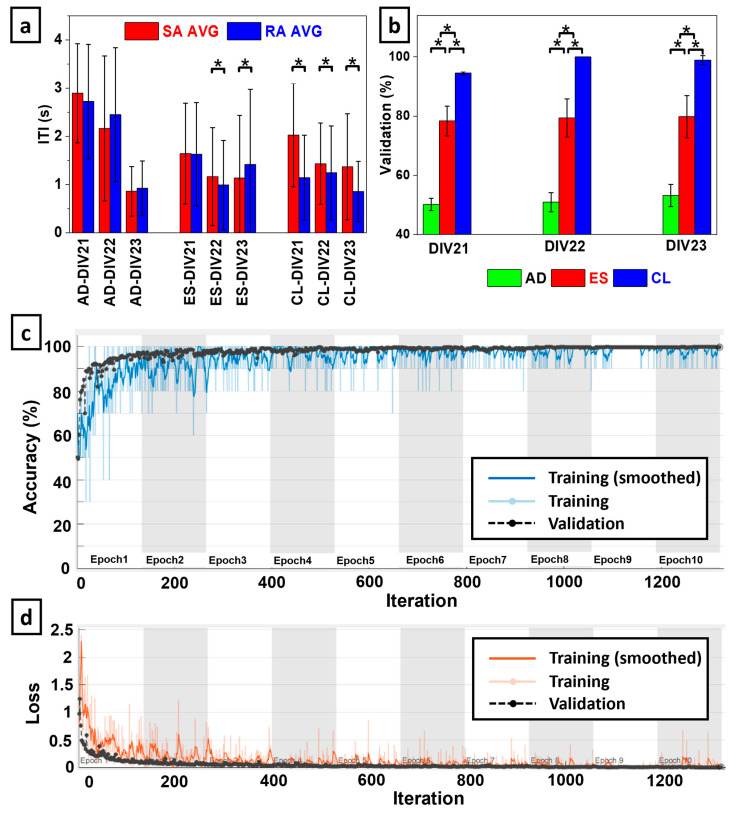
(**a**) Inter-Tap-Interval (ITI) comparison for RA and SA finger-tapping experiments from DIV21-DIV23 for Afferent-Deprived (AD), Efferent Substitution (ES), and Closed-Loop (CL) embodiments. Data showed functional specialization for ITI rates during the CL configuration. (**b**) CNN accuracies to classify mechanoreceptor encoding method with the AD, ES, and CL embodiment configurations from DIV21-DIV23. (**c**) Illustrative training progress with CL data and (**d**) corresponding training loss. * indicates statistical significance (*p* < 0.05).

**Table 1 biomimetics-09-00078-t001:** Izhikevich model parameters.

*a*	*b*	*c*	*d*	*X*	*Y*	*Z*
0.1	0.2	−65 mV	8	0.04	5	140

## Data Availability

The raw data supporting the conclusions of this article will be made available by the authors on request.

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
