# Peer review of "Biohybrid Robotic Hand to Investigate Tactile Encoding and Sensorimotor Integration"

_biomimetics, 2024, doi:10.3390/biomimetics9020078_

Round 1

Reviewer 1 Report (Previous Reviewer 2)

Comments and Suggestions for Authors

The paper has been improved and all my comments have been addressed. Just one suggestion: The paper looks very long for it's content. The authors may look to make it a bit more concise.

Author Response

Thanks for this comment. We have deleted repetitious statements throughout the manuscript to shorten the length of the paper. We deleted section 2.9, which was repetitious, and streamlined sentences throughout the paper to flow more smoothly. We also deleted several paragraphs in the Discussion that were similar to those in the Introduction. Finally, we shortened some of the lengthier figure captions. These changes have shortened the paper length and made it more readable.

Reviewer 2 Report (New Reviewer)

Comments and Suggestions for Authors

Comment on the originality:

A new biohybrid neuroprosthetic research platform is developed in the article. From a scientific point of view, it has the potential to become a useful clinical tool for studying optimal methods for neurorehabilitation with assistive robots.

Comments:

1. The title of the figure is too long and contains too much content. You can include a specific analysis of the figure in the main text.

2. In line 233, the PID controller is used in the article to control the change of finger joint angle, and the purpose and significance of PID control can be described more accurately. (Regarding the idea of a current PID controller.)

3. In line 59, it may be helpful to clarify the main differences between invasive and non-invasive neural interfaces.

4. Unnecessary spaces should be deleted from the formula (9). In line 472, the expression that triggers a spike is unclear and should be written (S, Figure 3.1(b)).

5. "FAC" and "FDC" are not uniformly represented in the article, either in italics or not. The same applies to "in vitro".

6. The lack of uniformity in the expression "Figure" is a difference between the full name and the abbreviation. In line 319, the expression "minutes" is the same.

7. In line 282, it may be helpful to clarify the work in "Python 3".

8. In line 519, how was "p<0.05" obtained and what does it mean?

9. In line 524, it would be useful to clarify the reason why the tactile sensation encoding method resulted in no consistent trend in ITI in the ES sessions. In line 530, the difference on DIV22 in the AD sessions is more significant and should be explained accordingly.

10.It should be clarified what the meanings of the different colored curves shown in Figures 6 (c) and 6 (d) are.

Comments on the Quality of English Language

The quality of the paper should be better in terms of language.

Author Response

Reviewer 3 Report (New Reviewer)

Comments and Suggestions for Authors

The paper is well orgnized and well written, the work is good, and I think it can be published.

Author Response

Thank you for your review of our paper.

This manuscript is a resubmission of an earlier submission. The following is a list of the peer review reports and author responses from that submission.

Round 1

Reviewer 1 Report

Comments and Suggestions for Authors

The study presents an experimental setup where a prosthetic hand is controlled via a biologically neural network (BNN) in a closed-loop fashion. Across different feedback conditions, a difference in the coherence across the input and output of the BNN (sensory feedback and motor commands, respectively) is reported. While the experimental setup is fascinating and the manuscript is written in a clear way, I must admit that in spite of my best efforts, I fail to understand the relevance of this model in the context of providing bidirectional prosthetic interfaces. The introduction explains the nature of the problem (lack of understanding of the impact of different feedback encoding schemes) but provides no clear justification for why exploiting BNN may provide relevant information that will help solve this problem. In spite of a rather lengthy introduction, the aim of the project is essentially described in a single sentence: “The goal of this paper is to explore the potential of a novel biohybrid neuroprosthetic hand to serve as a model to investigate the interaction between motor control of the hand and different electrical stimulation encoding methods for tactile sensations”. This sentence is not justified or elaborated, but simply followed by a description of the approach. What is missing, in my opinion, is a precise and well-founded hypothesis that states an expected outcome (e.g., the classification accuracy of the ANN) and, importantly, the potential implications of this outcome. For example, the state of the art within BNN (lines 78-91) is not described in the relevant context. Therefore, it is not clear to me how the experimental setup is related to the human-in-the-loop setup of normal prosthesis use. Similarly, the results indicate that the different experimental conditions imply different coherence spectra, but how can these differences help researchers in their efforts to understand the best feedback encoding scheme? I understand fully that this is the first study of its kind, and that an outcome that provides one clear answer to this question is not required. However, the authors must be able to explain how this novel experimental model may eventually lead to a step forward in the development of bidirectional prosthetic interfaces. I don’t see a clear explanation of these critical points in the introduction or in the discussion. For this reason, the study seems more like an exercise in applying BNN in a somewhat randomly selected context.

Reviewer 2 Report

Comments and Suggestions for Authors

Please find the comments in the attached file.

Round 2

Reviewer 1 Report

Comments and Suggestions for Authors

I regret to say that my concerns from the first round of comments remain. In the revised text, the authors state that the findings "could have important implications for amputees and SCI victims". In its essence, my concern comes down to a failure to understand these implications. It is certainly correct that "Dexterous grasp control is not yet completely understood in healthy people", but this is a very simplified statement. As clear from the reference for this statement (65), however, many aspects about such control is understood and it is unclear how the proposed setup can help clarify specific unclear aspects. In addition, it is unclear how clarifying these specific aspects may benifit prosthetic control. I acknowledge that the authors have included two hypotheses in the manuscript, but I fail to see a clear justification and most importantly, the implications for the research field if these hypotheses are confirmed. I respect that there should be room for introducing completely new methods into a research field, and doing so inevitably involves some uncertainty. However, there must be a clear overarching hypothesis indicating the potential relevance.